# Effect of Magnesium Salt (MgCl_2_ and MgSO_4_) on the Microstructures and Properties of Ground Granulated Blast Furnace Slag (GGBFS)-Based Geopolymer

**DOI:** 10.3390/ma15144911

**Published:** 2022-07-14

**Authors:** Kun Zhang, Kaiqiang Wang, Zhimao Liu, Zhiwu Ye, Baifa Zhang, Deng Lu, Yi Liu, Lijuan Li, Zhe Xiong

**Affiliations:** 1China Construction Third Engineering Bureau Group Co., Ltd., Wuhan 430074, China; zhangk@cscec.com (K.Z.); wangkaiqiang@cscec.com (K.W.); lzm_24@163.com (Z.L.); yzw1512@cscec.com (Z.Y.); 2School of Civil and Transportation Engineering, Guangdong University of Technology, Guangzhou 510006, China; 15374213357@163.com (Y.L.); lilj@gdut.edu.cn (L.L.); gdgyxz263@gdut.edu.cn (Z.X.)

**Keywords:** alkaline solution, geopolymer, ground granulated blast furnace slag, magnesium salt, microstructure

## Abstract

The use of seawater to prepare geopolymers has attracted significant research attention; however, the ions in seawater considerably influence the properties of the resulting geopolymers. This study investigated the effects of magnesium salts and alkaline solutions on the microstructure and properties of ground-granulated-blast-furnace-slag-based geopolymers. The magnesium salt–free Na_2_SiO_4_-activatied geopolymer exhibited a much higher 28 d compressive strength (63.5 MPa) than the salt-free NaOH-activatied geopolymer (31.4 MPa), with the former mainly containing an amorphous phase (C-(A)-S-H gel) and the latter containing numerous crystals. MgCl_2_·6H_2_O addition prolonged the setting times and induced halite and Cl-hydrotalcite formation. Moreover, mercury intrusion porosimetry and scanning electron microscopy revealed that the Na_2_SiO_4_-activated geopolymer containing 8.5 wt% MgCl_2_·6H_2_O exhibited a higher critical pore size (1624 nm) and consequently, a lower 28 d compressive strength (30.1 MPa) and a more loosely bound geopolymer matrix than the salt-free geopolymer. In contrast, MgSO_4_ addition had less pronounced effects on the setting time, mineral phase, and morphology. The Na_2_SiO_4_-activated geopolymer with 9.0 wt% MgSO_4_ exhibited a compressive strength of 42.8 MPa, also lower than that of the salt-free geopolymer. The results indicate that Cl^−^ is more harmful to the GGBFS-based geopolymer properties and microstructure than SO_4_^2−^ is.

## 1. Introduction

Alkali-activated geopolymers are inorganic cementitious materials composed of crosslinked [AlO_4_] and [SiO_4_] tetrahedrons, within charge-balancing hydrated alkali metal cations [1,2]. Upon the activation of a geopolymer precursor with a concentrated alkaline solution, it undergoes dissolution, rearrangement, condensation, and re-solidification [3]. Finally, a geopolymer with a three-dimensional network structure is formed after curing at ambient or slightly elevated temperatures [4]. Geopolymers exhibit excellent characteristics, such as favorable mechanical properties [5], durability [6], heavy metal solidification [7,8], and absorption [9]. In addition, their manufacturing process consumes less energy and emits less CO_2_ than the ordinary Portland cement (OPC) manufacturing process; thus, geopolymers have attracted considerable research attention and have been regarded as a potential alternative to OPC for preparing concrete materials [10,11,12]. Raw materials with sufficient amounts of reactive alumina and silica (such as clay minerals [13,14], solid waste [15,16,17], and other minerals [18,19]) and solid wastes such as mine tailings [17,20], fly ash (FA) [21,22], and slag [23,24] can be directly used as geopolymer precursors. In particular, finding use for solid waste in the concrete industry promotes environmental protection.

Ground granulated blast furnace slag (GGBFS) is a solid waste produced at a large scale by the iron industry. To efficiently recycle GGBFS, its applications as an additive, as a microcrystalline glass component, and as an alternative to cement have been explored in the fields of construction and medicine [25]. Owing to its high content of glassy phase and SiO_2_, CaO, and Al_2_O, GGBFS is a promising geopolymer precursor [26]. Thus far, GGBFS has been widely used as a raw material for geopolymer preparation and application in high-performance concrete [27], infrastructure construction [28], and composite manufacturing [29]. In India, Tembhurkar et al. [28] produced a modular toilet unit using GGBFS/FA-based geopolymer. They found that compared with traditional construction materials, GGBFS/FA-based geopolymer conserved natural resources, reduced CO_2_ emissions, and reduced production costs.

Concrete production consumes not only cementitious materials but also a large amount of natural resources such as fresh water, which can contribute to water shortages. Annually, approximately 1.5 billion tons of fresh water is consumed for concrete production; by 2025, two-thirds of the world population is estimated to face a water crisis [30], exacerbated by the uneven distribution of water resources [31]. With the increased demand for environmental conservation, finding an alternative to fresh water for use in construction is vital.

Construction projects on islands and offshore sites have largely increased. Therefore, the use of seawater instead of fresh water to prepare concrete has attracted increasing attention [32,33,34]. Wang et al. [35] found that seawater could enhance OPC hydration, resulting in microstructure densification, high crystal content, and reduced autogenous shrinkage. Several researchers have recently utilized seawater for geopolymer preparation. Rashad et al. [36] compared the effects of magnetic water, seawater, and tap water on the workability and mechanical properties of slag-based geopolymer and found that seawater, owing to its high pH and high amounts of Cl^−^, Na^+^, and SO_4_^2−^, improved the flowability and compressive strength of the pastes. Jun et al. [37] comparatively studied the properties of geopolymers prepared with seawater and deionized water and concluded that seawater improved electrical resistivity and early compressive strength, attributable to the formation of Cl-bearing phases (e.g., Cl-hydrocalumite, AlOCl, and aluminum chloride hydrate). Moreover, the seawater-mixed, slag-based geopolymer exhibited a higher Cl-binding capacity than the FA-based geopolymer, owing to the differences in the Cl-bearing composition [38].

The findings of the above-mentioned studies indicate that seawater is a potential material for use in preparing alkali-activated geopolymer; however, the ions (e.g., Cl^−^, Mg^2+^, and SO_4_^2−^) in seawater affect the geopolymer microstructure and chemical composition, thereby altering the geopolymer mechanical properties and durability. Several studies [39,40,41] have clarified the role of seawater in geopolymer preparation and confirmed that the microstructures of the as-obtained materials highly influenced their macro-properties. While numerous studies have focused on the comparative evaluation of the mechanical properties of seawater-mixed and deionized water-mixed geopolymers and on the role of Cl^−^ in geopolymers [37,38,42,43,44], few studies have examined the effects of other ions on geopolymer microstructures and properties. In particular, SO_4_^2−^ may have a remarkable impact on the microstructures and properties of slag-based geopolymers [45]. Studies have reported that in geopolymers immersed in sulfate solution or seawater, SO_4_^2−^ reacts with Ca^2+^ to form expandable gypsum and ettringite [46], destroying the geopolymer microstructure. These studies have also reported that Ca considerably influences the composition and properties of salt-containing, water-mixed geopolymer. Thus far, a few studies have shown that Cl-containing salts (e.g., KCl, MgCl_2_, and CaCl_2_) induce the precipitation and crystallization of geopolymeric gels, whereas CO_3_^2−^-containing salt (e.g., K_2_CO_3_, MgCO_3_, and CaCO_3_) prevents hydrolytic attacks on geopolymeric gels [47]. In other studies, SO_4_^2−^ facilitates zeolite crystallization in geopolymers [48], and high SO_4_^2−^ content is much more beneficial to geopolymer properties than low SO_4_^2−^ content [49]. Thus, the anions considerably influence the geopolymer properties. However, the geopolymers prepared in the above-mentioned studies were based on FA or kaolinite, both of which have relatively low Ca concentrations.

In this work, considering that seawater mainly contains Na^+^, Mg^2+^, Cl^−^, and SO_4_^2−^ ions while alkaline solutions contain large amounts of Na^+^, the effects of the salt (MgCl_2_, MgSO_4_) content on the mechanical properties and microstructures of alkali-activated (NaOH or Na_2_SiO_3_ solution) GGBFS-based geopolymers were investigated, with the objective of elucidating the geopolymerization of Ca-containing precursors in salt-containing seawater. The chemical structures of different polymers were characterized via X-ray diffraction (XRD), Fourier-transform infrared (FTIR) spectroscopy, and thermogravimetric (TG) analysis. The microstructural features and composition of the geopolymers were examined via scanning electron microscopy (SEM) coupled with energy-dispersive X-ray (EDX) spectroscopy and mercury intrusion porosimetry (MIP).

## 2. Materials and Methods

### 2.1. Materials

GGBFS was purchased from Yuanheng Environmental Protection Engineering Co., Ltd., in Henan province, Nanyang, China. Its chemical composition (determined via X-ray fluorescence) by mass of oxides was as follows: SiO_2_ (33.21 wt%), CaO (37.14 wt%), Al_2_O_3_ (15.76 wt%), Fe_2_O_3_ (0.71 wt%), MgO (8.51 wt%), TiO_2_ (1.91 wt%), and others (2.76 wt%). CaO and SiO_2_ accounted for 70.35 wt% of the total GGBFS composition. The XRD patterns (Figure 1a) featured one broad peak spanning from 19°(2θ) to 40°(2θ), indicating that GGBFS was mainly in the amorphous phase, with small amounts of crystals, such as calcite, akermanite, and dolomite. SEM images (Figure 1b) showed that the particles in the GGBFS were nonuniform in size and angular.

Chemical-grade NaOH pellets (Zhiyuan Chemical Reagent Co., Ltd., Tianjin, China; purity ≥ 96%) and Na_2_SiO_4_ pellets (Henyuan New Material Co., Ltd., Zhengzhou, China; Na_2_O 26.0 wt% and SiO_2_ 60.0 wt%, modulus of 2.9) were used as alkali activators. MgCl_2_·6H_2_O (Yongda Chemical Reagent Co., Ltd., Tianjin, China; purity ≥ 98%) and MgSO_4_ (Baishi Chemical Co., Ltd., Tianjin, China; purity ≥ 99%) were used as additives. All of the chemical agents were obtained from commercial suppliers.

### 2.2. Geopolymer Preparation

First, NaOH or Na_2_SiO_4_ was added into distilled water to prepare an alkaline solution. The NaOH concentration was set to 10 mol/L, while the Na_2_SiO_4_ concentration was set to 35%, with a modulus of 1.5. The desired concentrations were achieved through the controlled addition of NaOH, Na_2_SiO_4_ pellets, and deionized water. As shown in Figure 2, GGBFS was mixed with an alkaline solution at a solid/liquid ratio of 0.55 to form a homogeneous paste, which was then cast into silica molds (20 × 20 × 20 mm^3^). To prevent water evaporation, the molds were covered with a thin polyethylene film, and the molded specimens were cured at ambient temperature for 24 h before demolding. The demolded specimens were cured at ambient temperature until the test date.

The concentration of MgSO_4_ salt in GGBFS was set to 1.0, 3.0, 5.0, 7.0, and 9.0 wt%, while, based on the number of Mg^2+^ moles in MgSO_4_, the MgCl_2_·6H_2_O concentration in GGBFS was set to 1.7, 5.1, 8.5, 11.8, and 15.2 wt%. The salts were first ground into powder and then mixed with GGBFS; afterward, NaOH or Na_2_SiO_4_ solution was added, and the mixture was stirred to form a homogeneous paste. The preparation process and conditions were the same as those for the activated GGBFS sample, except for the specimens containing 11.8 and 15.2 wt% MgCl_2_·6H_2_O. These two specimens could not set at ambient temperature and thus were cured at 40 °C in an oven. Herein, the specimens are denoted as G_X_-Cl_Y_ or G_X_-S_Y_, where X represents the modulus of the alkaline solution and Y the salt concentration by weight, and Cl and S represent MgCl_2_·6H_2_O and MgSO_4_, respectively. For example, G_0_-Cl_1.7_ denotes a GGBFS-based geopolymer prepared via activation with a 10 mol/L NaOH solution and containing 1.7 wt% MgCl_2_·6H_2_O.

### 2.3. Characterization Methods

According to the ASTM C191 standard test method, the initial and final setting times of the alkali-activated, GGBFS-based geopolymeric pastes were determined using Vicat apparatus (Luda Experimental Instruments Co., Ltd., Shanghai, China).

According to the GBT50081-2019 standard test method, the compressive strengths of all geopolymers cured for 1, 7, and 28 days were tested on a microcomputer-controlled electronic universal testing machine (STS100K, Yishite Instrument Co., Ltd., Xiamen, China) under a loading rate of 100 N/s. The compressive test for each group was run four times, and the average value was recorded as the compressive strength.

The XRD patterns, in the 3°(2θ)–60°(2θ) range, of the selected geopolymers cured for 28 days were obtained using a diffractometer (Bruker D8 Advance, Mannheim, Germany) with CuKα radiation operated at 40 kV, 40 mA, and a scanning speed of 3°(2θ)/min.

The FTIR spectra (64 scans, 4 cm^−1^) of selected geopolymers cured for 28 days were obtained using a spectrometer (Nicolet IS50, Thermo Fisher, Waltham, MA, USA) at a wavelength of 400–1800 cm^−1^. First, 0.8 mg of each sample was mixed with 80 mg of KBr, and then, the mixture was ground into fine powder. The mixture was then pressed into a disk for testing.

The TG and differential scanning calorimetry (DSC) curves of selected geopolymers cured for 28 days were obtained using a thermal gravimetric analyzer (TGA 4000, PE, Holland). The powdered samples were heated between 30 °C and 900 °C at 10 °C/min.

The SEM micrographs and EDX spectra of selected geopolymers cured for 28 days were obtained using a scanning electron microscope (S-3400N-II, Hitachi, Tokyo, Japan) at an accelerating voltage of 15 kV. The samples were anchored on a conducting tape and then coated with a gold layer.

The pore size distribution of selected geopolymers was determined using a mercury porosimeter (AutoPore Iv 9510, Micromeritics, Norcross, GA, USA). The intrusion pressures ranged from 0.5 to 33,000 psi, and the equilibrium time for each applied pressure was set to 10 s.

## 3. Results and Discussion

### 3.1. Setting Time and Compressive Strength

#### 3.1.1. Setting Time

Figure 3 shows the initial and final setting times of the geopolymers. The initial and final setting times of NaOH-activated GGBFS were slightly longer than those of Na_2_SiO_3_-activated GGBFS, demonstrating that the presence of soluble silicate favored the setting and hardening of geopolymeric pastes.

The setting times of NaOH-activated GGBFS when increasing MgCl_2_·6H_2_O from 0 to 15.2 wt% decreased slightly from 20 to 12 min and then increased to 18 min. Meanwhile, the setting time of geopolymer (G_0_-0) decreased gradually from approximately 20 min to 10 min when the addition dosage of MgSO_4_ increased from 0 to 9 wt%. Thus, the addition of a small amount of magnesium salt shortened the setting time of NaOH-activated GGBFS, possibly owing to the exothermic reaction between Mg^2+^ and OH^−^. However, large amounts of OH^−^ were consumed at a high dosage of magnesium salt, which was accompanied by alkalinity reduction and the extension of the geopolymer setting time.

Figure 3b shows that the final setting time of the Na_2_SiO_3_-activated GGBFS with 5.0 wt% MgSO_4_ (9 min) was shorter than that of the salt-free, Na_2_SiO_3_-activated GGBFS (G_1.5_-0, 20 min), but the geopolymer with 9 wt% MgSO_4_ exhibited a significantly longer setting time (41 min). In particular, at a MgCl_2_·6H_2_O dosage of >8.5 wt%, the geopolymer paste could not set at the ambient temperature. At high magnesium salt concentrations, the occurrence of Cl^−^ and SO_4_^2^^−^ prolonged the setting time of Na_2_SiO_3_-activated GGBFS relative to that of NaOH-activated GGBFS (Figure 3a).

In addition, the Mg^2+^ anion considerably influenced the setting behavior of the GGBFS-based geopolymer. At high concentrations, Cl^−^ tended to prolong the geopolymer setting time more than SO_4_. This result is consistent with those of previous studies, which have reported that a high Cl^−^ concentration severely hinders the setting of GGBFS-based geopolymers [50]; however, SO_4_^2^^−^ exhibited a less pronounced influence on the setting time of Na_2_SiO_3_-activated, high-calcium, FA-based geopolymers [51].

#### 3.1.2. Compressive Strength

Figure 4 shows the compressive strengths of all the tested geopolymers. The compressive strengths increased with aging time. In addition, the NaOH-activated GGBFS showed much lower compressive strengths than the Na_2_SiO_3_-activated GGBFS. The 28 d compressive strength of G_1.5_-0 was 63.5 MPa, while that of G_0_-0 was only 31.4 MPa. Studies have also reported that the mechanical properties of geopolymers considerably improve with an increasing Si/Al ratio [52,53].

Figure 4a shows that magnesium salt addition negatively affected the compressive strengths of NaOH-activated GGBFS. With an increase in the MgCl_2_·6H_2_O concentration from 0 to 8.5 wt%, the 28 d compressive strength of the geopolymer decreased from 31.4 to 14.7 MPa; however, with a further increase to 15.2 wt%, the compressive strength barely changed. The G_0_-S samples exhibited similar results: the 28 d compressive strengths of the geopolymer decreased from 31.4 to 23.8 MPa with an increase in the magnesium salt concentration from 0 to 5 wt% and then remained almost unchanged with a further increase. These results suggest that at a high magnesium salt concentration, the anions offset the negative effect of Mg^2+^, thereby improving the compressive strength of the NaOH-activated, GGBFS-based geopolymer.

The addition of magnesium salts also reduced the compressive strength of Na_2_SiO_3_-activated GGBFS (Figure 4b), with Cl^−^ exhibiting a greater negative effect than SO_4_^2^^−^. With an increase in the MgCl_2_·6H_2_O concentration from 0 to 8.5 wt%, the 1 d and 28 d compressive strengths of the geopolymer decreased significantly from 50.5 to 0.85 MPa and from 63.5 to 30.1 MPa, respectively. Geopolymers with higher MgCl_2_·6H_2_O concentrations could not set at the ambient temperature. Moreover, increasing the curing temperature improved geopolymerization; accordingly, G_0_-Cl_11.8_ and G_0_-Cl_15.2_ exhibited 28 d compressive strengths of 41.9 and 22.1 MPa, respectively. The geopolymer with a MgSO_4_ concentration of 9.0 wt% exhibited a 28 d compressive strength of 42.8 MPa, less than that of the geopolymer with 8.5 wt% MgCl_2_·6H_2_O (63.5 MPa).

### 3.2. Structure of GGBFS-Based Geopolymer

#### 3.2.1. XRD Patterns

Figure 5 shows the XRD patterns of selected GGBFS-based geopolymers. G_0_-0 showed a different diffraction pattern from G_1.5_-0. The XRD pattern of G_0_-0 featured a broad peak ranging from 24°(2θ) to 42°(2θ). The peak was centered at 32°(2θ) in the XRD patterns of the geopolymer but at 30°(2θ) in those of GGBFS (Figure 1), indicating the formation of an amorphous geopolymer (C-(A)-S-H), which may contribute to strength development. In addition to the amorphous phase, many neoformed crystals occurred in the geopolymers. Peaks corresponding to crystal C-(A)-S-H and hydrotalcite-like (Mg_6_Al_2_(CO_3_)(OH)_16_·4H_2_O), a Mg-layered double hydroxide phase, were identified at approximately 29°(2θ) (with a *d*_022_ of 3.1 Å) and 11°(2θ) (with a *d*_006_ of 7.6 Å), respectively [54,55]. Chlormagaluminite ((Mg, Fe)_4_Al_2_(Cl, CO_3_)(OH)_12_·2H_2_O), with a comparable composition and structure with hydrotalcite-like phases, was also formed. Moreover, numerous unreacted species, such as calcite and akermanite, also occurred, as they did not react with NaOH. The magnesium-salt-containing geopolymers exhibited no considerable diffraction difference; they only differed in the intensity of XRD peaks corresponding to hydrotalcite-like phases, which was attributable to the replacement of CO_3_^2^^−^ by Cl^−^ or SO_4_^2^^−^, because the CO_3_^2^^−^ in the interlayers of the hydrotalcite-like phases was displaceable [38].

The XRD patterns of G_1.5_-0 showed a broad peak centered at approximately 29°(2θ) (Figure 4b). Although this center shifted to a lower region after geopolymerization, the geopolymer could be concluded to have formed [56]. In addition, G_1.5_-0 contained crystal C-(A)-S-H but no other mineral, unlike G_0_-0. Therefore, the soluble silica significantly affected the geopolymerization process and the chemical composition of alkali-activated GGBFS. The XRD pattern of the geopolymer with 9 wt% MgSO_4_ (G_1.5_-S_9_) was not considerably different from that of G_1.5_-0, indicating that Ca^2+^ preferentially reacted with silicate to form C-(A)-S-H gels than with SO_4_^2^^−^ to form gypsum (CaSO_4_). However, the determination of the occurrence form of SO_4_^2^^−^ via an XRD test is difficult because no SO_4_^2^^−^-containing mineral was detected. The XRD pattern of the geopolymer with 1.7 wt% MgCl_2_·6H_2_O (G_1.5_-Cl_1.7_) was the same as that of G_1.5_-0. However, the geopolymers containing 8.5 wt% MgCl_2_·6H_2_O contained halite (NaCl), and with a further increase in the salt concentration, the peaks corresponding to halite intensified, indicating the formation of more halite, which was attributable to the combination of Na^+^ and Cl^−^ during the curing process. These results suggest that Cl^−^ more easily reacted with Na^+^ than SO_4_^2^^−^ did because no crystal Na_2_SO_4_ was detected.

#### 3.2.2. FTIR Spectra

Figure 6 shows the FTIR spectra of selected geopolymers in the 400–1800 cm^−1^ range. The FTIR spectra of G_0_-0 (Figure 6a) featured an absorption band at approximately 1648 cm^−1^, corresponding to the bending vibration of H–O–H, resulting from water absorption [57]. The broad band at approximately 1452 cm^−1^ corresponded to the asymmetric stretching vibration of O–C–O from hydrotalcite-like phases, Na_2_CO_3_, and unreacted CaCO_3_ [55,58]. In addition, another band corresponding to the stretching vibration of carbonate from calcite occurred at 896 cm^−1^ [55]. However, this band has been reported to correspond to the stretching vibration of Si-O from C-(A)-S-H [58]. Moreover, bands between 1200 and 900 cm^−1^ corresponded to the asymmetric stretching vibration of Si-O-T (T: Si or Al); these bands, denoted as “main bands,” have been widely reported in research on the chemical properties of alkali-based geopolymers [59]. The small peak at 718 cm^−1^ corresponded to Si-O-Al^IV^ bending vibration, while those at 664 and 602 cm^−1^ corresponded to deformation vibrations of Si-O-T from C-(A)-S-H [54]. Furthermore, the absorption bands at 483 and 456 cm^−1^ corresponded to the bending vibration of Si-O-Si. After the addition of magnesium salts, the bands at 1452 cm^−1^ shifted to a slightly higher field, which is attributable to the displacement of CO_3_^2^^−^ and anions (i.e., Cl^−^ and SO_4_^2^^−^). The FTIR spectra featured no other change, indicating that the ions did not significantly alter the geopolymer molecular structure.

G_1.5_-0 exhibited fewer and broader FTIR peaks than G_0_-0 (Figure 6b), indicating that a higher amount of amorphous phase was formed in Na_2_SiO_3_-activated GGBFS than in NaOH-activated GGBFS. In addition, the main band of G_1.5_-0 (969 cm^−1^) was higher than that of G_0_-0 (953 cm^−1^), demonstrating the incorporation of more Si in G_1.5_-0 than in G_0_-0 [60,61]. The FTIR spectra of the NaOH-activated GGBFS exhibited a band at approximately 876 cm^−1^, attributable to the presence of natural bond orbitals (e.g., Al-O^−^, Si-O^−^) resulting from the geopolymerization process [13]. The FTIR spectra of the magnesium-salt-containing geopolymers mainly differed from that of the salt-free GGBFS in the positions of the main bands; that is, the main band slightly shifted to a higher wavenumber after magnesium salt addition, attributable to the occurrence of more unreacted GGBFS owing to the reduced alkalinity resulting from the reaction between alkali and magnesium ions [62,63]. Consequently, the compressive strength decreased with increasing salt concentration. In addition, the absorption bands at approximately 619 cm^−1^ in the G_1.5_-S_5_ and G_1.5_-S_9_ spectra could be attributed to the bending vibration of SO_4_^2^^−^ [64]. Moreover, the band at 1104 cm^−1^ in the G_1.5_-S_9_ spectrum corresponded to the asymmetric stretching of SO_4_^2^^−^ [65].

#### 3.2.3. TG Result

The thermal properties of G_0_-0, G_1.5_-0, G_1.5_-Cl_8.5_, G_1.5_-Cl_15.2_, and G_1.5_-S_9_ are shown in Figure 7. G_0_-0 exhibited a total mass loss of ~30%, which was much greater than that of G_1.5_-0 (~20%). The magnesium-salt-containing geopolymers exhibited a higher total mass loss, demonstrating that the salt restricted water evaporation from the geopolymeric paste. The Cl-containing geopolymers appeared to retain more water than the SO_4_^2^^−^-containing geopolymers because G_1.5_-Cl_8.5_ and G_1.5_-Cl_15.2_ exhibited a higher mass loss than G_1.5_-S_9_ after calcination to 900 °C.

The DSC curves of the geopolymers featured an endothermic peak ranging from 30 °C to 150 °C, attributable to the evaporation of free water and the decomposition of C-(A)-S-H gels [15]. Accordingly, considerable mass loss (>75%) occurred at this temperature range. The Cl-containing geopolymers exhibited the highest mass loss before 150 °C, presumably because they retained the highest amount of free water. The continuous mass loss during calcination between 150 °C and 300 °C for all geopolymers can be attributed to the loss of bound water (interstitial water). G_0_-0 retained the most interstitial water [66]. Furthermore, the mass loss during calcination at 300–600 °C was mainly due to the decomposition of hydroxyl water (dehydroxylation) [66]. The DSC curve of G_0_-0 featured a small endothermic peak, attributable to hydrocalcite decomposition; this peak was nonexistent in other samples [55]. These results substantiate the formation of hydrocalcite in NaOH-activated GGBFS-based geopolymers but not in Na_2_SiO_4_-activated geopolymers. Accordingly, G_0_-0 exhibited a higher mass loss than the other samples at 300–600 °C. In addition, at temperatures exceeding 600 °C, the geopolymers exhibited little change in mass loss, consistent with the findings of previous studies [4,67].

### 3.3. Microstructure of GGBFS-Based Geopolymers

#### 3.3.1. SEM/EDX Result

Figure 8, Figure 9 and Figure 10 show the SEM images of selected geopolymers, and Figure 11 displays the EDX spectra of the red spots in Figure 8, Figure 9 and Figure 10. G_0_-0 exhibited a compact microstructure but also had some cracks and macropores (Figure 8a). The gels were connected firmly, but many isolated particles still occurred, resulting in microstructure inhomogeneity. The EDX result (Figure 11, spot 1) indicated that the G_0_-0 matrix was mainly composed of O, Na, Ca, Si, and Al, indicating the formation of N,C-(A)-S-H gels and a strengthening effect. After the addition of magnesium salts, the geopolymer matrix became loosely bound, accompanied by the development of numerous cracks and pores (Figure 8c,e). However, the EDX results of G_0_-Cl_15.2_ (Figure 11, spot 2) indicated that the Cl^−^ ions were not evenly distributed in the matrix. These results indicate that Cl^−^ may react with Na to form halite or displace CO_3_^2^^−^ to form Cl-hydrotalcite, rather than spread in the matrix of the NaOH-activated, GGBFS-based geopolymer. G_0_-S_9_ exhibited a similar result (Figure 11, spot 3); its S concentration was substantially less than that of G_1.5_-S_9_ (Figure 11, spot 7). Although Mg^2+^ was also detected, GGBFS inherently contained some MgO; thus, it is difficult to determine the role of Mg^2+^. In addition, some plate- and needle-like crystals occurred in G_0_-Cl_15.2_ (Figure 8d); the plate-like crystals, owing to their hexagonal shape, could be Friedel’s salt from the Cl^−^ content. Figure 8f shows that numerous large voids occurred in G_0_-S_9_, which reduced the ultimate compressive strength.

Figure 9 displays the SEM images of G_1.5_-0, G_1.5_-S_1_, and G_1.5_-S_9_. G_1.5_-0 exhibited a considerably more compact morphology than G_0_-0. Particularly, G_1.6_-0 exhibited a more homogeneous and denser matrix than G_0_-0 (Figure 8b and Figure 9b), corresponding to a higher compressive strength. However, numerous microcracks occurred in the matrix. Numerous microcracks also occurred in G_1.5_-S_1_ and G_1.5_-Cl_9_ (Figure 9d,f). The EDX results showed that the G_1.6_-0 matrix (Figure 11, spot 4) also contained O, Na, Ca, Si, and Al to form N,C-(A)-S-H gels. In addition, G_1.5_-0 exhibited a higher Si concentration than G_0_-0, which was also related to the higher compressive strength of G_1.5_-0 than G_0_-0 [53]. Furthermore, the homogeneous-matrix geopolymer exhibited higher Ca and Si concentrations (Figure 11, spot 5) than the geopolymer with a less homogeneous matrix (Figure 11, spot 4), which substantiates the important roles of Ca and Si in the geopolymer matrix densification.

The morphologies of the MgSO_4_-containing geopolymers were not considerably different from that of the MgSO_4_-free geopolymer. G_1.5_-S_1_ exhibited a relatively compact and dense morphology (Figure 9c,e) but contained numerous microcracks (Figure 9d,f). In addition, the geopolymer matrix contained numerous macropores, which originated from the entrapped air. The S concentration of the MgSO_4_-containing geopolymers, detected via EDX spectroscopy, increased with increasing MgSO_4_ concentration (Figure 11, spots 6, 7), indicating that SO_4_^2^^−^ was evenly distributed in the geopolymer matrix. However, the Mg^2+^ concentration did not change in this manner.

Figure 10 shows the SEM images of G_1.5_-Cl. G_1.5_-Cl_1.7_, which exhibited a dense microstructure (Figure 10a) and showed no morphological difference from G_1.5_-0. Only a little amount of Cl element was detected via EDX spectroscopy (Figure 11, spot 8). The geopolymer with a salt concentration of 8.5 wt% exhibited a loosely bound matrix, macrocracks, uneven surface, and an inhomogeneous microstructure (Figure 10c), and numerous particles were distributed on the surface of the geopolymer matrix (Figure 10d). G_1.5_-Cl_15.2_ exhibited a more compact matrix than G_1.5_-Cl_8.5_ (Figure 10e), indicating that increasing the curing temperature favored the densification of the geopolymer matrix. However, numerous pores and particles occurred on the matrix surface. All of these changes in the microstructure reduced compressive strength.

The EDX result showed that Cl^−^ was evenly distributed in the matrix (Figure 11, spots 9, 10). With increasing MgCl_2_·6H_2_O concentration, the Cl content increased, while the Ca concentration decreased and the Na concentration gradually increased, suggesting that Cl^−^ enhanced N-A-S-H gel formation.

#### 3.3.2. MIP Result

Figure 12 displays the pore size distribution and cumulative intrusion curve of GGBFS-based geopolymers, and Table 1 shows the pore parameters of the geopolymers. Pore structure is a major parameter influencing the mechanical properties of geopolymers. Wu et al. [68] categorized pores in cement into three groups according to pore size: gel pores (<10 nm), capillary pores (10–5000 nm), and macropores (>5000 nm).

Figure 12a illustrates the critical pore sizes of the salt-free and salt-containing Na_2_SiO_4_-activated geopolymers. G_1.5_-0 mostly exhibited capillary pores of ~132 nm. The MgCl_2_·6H_2_O-containing geopolymers exhibited larger capillary pores, as indicated by the higher peaks, and macropores of ~10,556 nm. These results demonstrate that MgCl_2_·6H_2_O enlarged the geopolymer pore structure. In addition, with increasing MgCl_2_·6H_2_O concentration, the total pore area of G_1.5_-Cl increased, and the average pore diameter gradually decreased (Table 1). G_1.5_-Cl_15.2_ exhibited few capillary pores of ~18 nm, as indicated by the sharp peak, attributable to halite formation. Figure 12b indicates that MgCl_2_·6H_2_O addition increased the geopolymer pore volume, except for G_1.5_-Cl_1.7_, consistent with the total pore area results (Table 1). Moreover, the porosity of G_1.5_-Cl increased with increasing MgCl_2_·6H_2_O concentration. Therefore, high MgCl_2_·6H_2_O concentrations deteriorate the pore structure of the geopolymer and thus its compressive strength.

MgSO_4_ addition also increased the critical pore size of the geopolymer. As the MgSO_4_ concentration increased from 0 to 9.0 wt%, the capillary pore size gradually increased from 132 to 2846 nm, and macropores progressively developed, as indicated by the corresponding more intense and broader peaks. Therefore, salt addition favored macropore production in the geopolymer. Studies have shown that pores of 10 to 1000 nm may be harmful to the mechanical properties of cement [69]. With increasing critical pore size, compressive strength reduced. Compared with G_1.5_-0, G_1.5_-S_5_ exhibited a lower pore volume, while G_1.5_-S_9_ exhibited basically the same pore volume, different from the case of G_1.5_-Cl. This was also evidenced by the geopolymer porosity values (Table 1). These results demonstrate that different ions had different effects on the geopolymer pore structure, attributable to the formation of different products. However, the related influence mechanism remains unclear, which necessitates further study. Although the pore volume decreased, the compressive strength also decreased, presumably because some pores were filled with SO_4_^2^^−^-containing precipitation and thus did not contribute to strength.

### 3.4. Summary and Final Discussion

This study found that both MgSO_4_ and MgCl_2_ negatively affected the microstructure and compressive strength of alkali-activated, GGBFS-based geopolymers but did not considerably affect the chemical composition. For example, the addition of 10 wt% MgSO_4_ did not alter the XRD patterns of G_1.5_-0 but reduced the 28 d compressive strength by 32.7%, while the addition of 8.5 wt% MgCl_2_ resulted in the formation of only halite in the geopolymer but reduced the 28 d compressive strength by 51.5%. These results are consistent with those of previous studies. Lee et al. [47] studied the effect of inorganic salt on the compressive strength of kaolin/FA-based geopolymer. The 270 d compressive strengths decreased from 65.6 MPa to 24.9, 18.4, and 12.2 MPa after the addition of 0.08 moles of KCl, CaCl_2_, and MgCl_2_, respectively. Criado et al. [48] investigated the effect of Na_2_SO_4_ on the alkali activation of an FA-based geopolymer and found that Na_2_SO_4_ addition reduced the 180 d compressive strength of the geopolymer from 78.0 to 40.5 MPa; they attributed this decrease to the drop in the N-A-S-H concentration after Na_2_SO_4_ addition.

However, in the current study, magnesium salt addition resulted in a more significant reduction in early compressive strength than late compressive strength. In contrast, previous studies on seawater-mixed geopolymers have shown that the ions in seawater significantly increase the early compressive strength of the geopolymer. Ren et al. [48] observed that the 1 d compressive strength of seawater-mixed, slag-based geopolymer was 27.2 MPa, slightly greater than that of the distilled-water-mixed geopolymer (23.6 MPa). Lv et al. [70] reported that one-part alkali-activated GGBFS-FA mixed with tap water and seawater exhibited comparable flexural and compressive strengths. However, the seawater-mixed geopolymer showed slightly higher early strengths (1 and 14 days) than the tap-water-mixed geopolymer. Kang et al. [71] reported that compared with tap water, seawater as mixing water reduced the large- and medium-sized pores and increased the gel pores of slag-based geopolymers; therefore, the seawater-mixed geopolymer showed higher 3 d compressive strength than the tap-water-mixed geopolymer. The discrepancy between the effects of salt ions between the current study and previous studies is attributable to the presence of multiple salts in seawater compared with the sole salt in this study; nonetheless, further study on this discrepancy is warranted.

This study revealed that the microstructure and mechanical properties of GGBFS-based geopolymers were remarkably influenced by magnesium salt. The findings have several implications for the application of alkali-activated, GGBFS-based geopolymers. For example, considering that Cl exhibited a more pronounced negative effect than SO_4_, practitioners should pay more attention to Cl during the preparation of GGBFS-based geopolymers with seawater or ion-contaminated water. In addition, the ions more significantly reduced the early mechanical properties than the late mechanical properties (Figure 4). Therefore, ion-contaminated, GGBFS-based geopolymers should be sufficiently cured to achieve high strength before application in infrastructure construction or composite manufacturing. Moreover, the ions present in the geopolymers can retain water (Figure 7), reducing the geopolymer’s propensity to shrink. The water retention property must be carefully considered before the application of ion-contaminated, GGBFS-based geopolymers in some dry areas to prevent the significant loss of water.

Some limitations also need to be considered. First, the leaching behavior of Mg^2+^, Cl^−^, and SO_4_^2^^−^ was not well established in the literature, and the solidification ability of ions for ion-contaminated, GGBFS-based geopolymers is vital to their applicability. For example, free Cl^−^ can initiate rebar corrosion; thus, materials with reduced free Cl^−^ are beneficial for rebar construction [72]. Second, geopolymer durability was not investigated in this study. Good durability, which is related to high anti-chloride penetration, anti-sulfate attack, and anti-carbonation, is essential in applications in marine environments [73]. Lastly, this study only investigated the composition, microstructure, and compressive strength of ion-contaminated mixed geopolymers aged for up to 28 days. With increasing aging time, the mineral composition and mechanical properties may significantly change. Burciaga-Díaz et al. [55] found that a NaOH-activated, slag/metakaolin-based geopolymer transitioned from an amorphous phase into a well-ordered phase over 6 years, with corresponding changes in physical and chemical properties. Therefore, the long-term properties of the investigated geopolymers need to be further explored.

## 4. Conclusions

In this study, through spectroscopic and microscopic techniques, we investigated the effects of magnesium salts, the main ion content of seawater, on the compressive strength and microstructure of as-obtained, GGBFS-based geopolymers.

The activation of GGBFS with NaOH resulted in the formation of C-(A)-S-H crystals, C-(A)-S-H gels, hydrotalcite-like phases, and chlormagaluminite. The addition of MgCl_2_·6H_2_O or MgSO_4_ loosened the microstructure of the geopolymer, thereby reducing its compressive strength. With increasing salt concentration, the compressive strength gradually decreased. MgCl_2_·6H_2_O addition induced the formation of halite and Cl-hydrotalcite, whereas MgSO_4_ addition did not significantly alter the mineral phase and morphology of the geopolymer.

The activation of GGBFS with Na_2_SiO_3_ mainly produced C-(A)-S-H crystal and amorphous C-(A)-S-H gels. The anions (Cl^−^, SO_4_^2^^−^) were evenly distributed in the geopolymer matrix after magnesium salt addition. With increasing magnesium salt concentration, the critical pore size of the geopolymer increased, the microstructure became less compact and loosely bound, and thus the compressive strength gradually decreased. MgCl_2_·6H_2_O addition induced halite formation and increased the geopolymer pore volume, whereas MgSO_4_ addition did not significantly alter the mineral phase and pore volume of the geopolymer.

Further research into the durability and long-term properties of seawater-mixed, GGBFS-based geopolymers, the role of Mg ions, and the role of other salts is still required to elucidate the practical applicability of the geopolymers.

## Figures and Tables

**Figure 1 materials-15-04911-f001:**
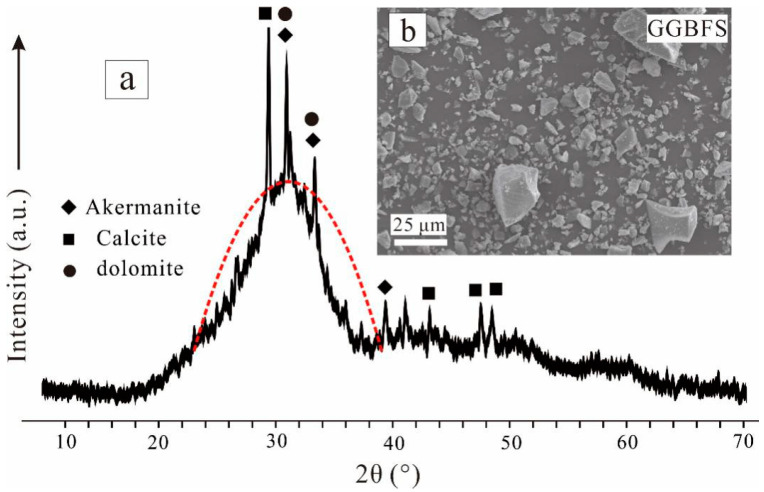
(**a**) XRD patterns and (**b**) SEM image of the GGBFS.

**Figure 2 materials-15-04911-f002:**
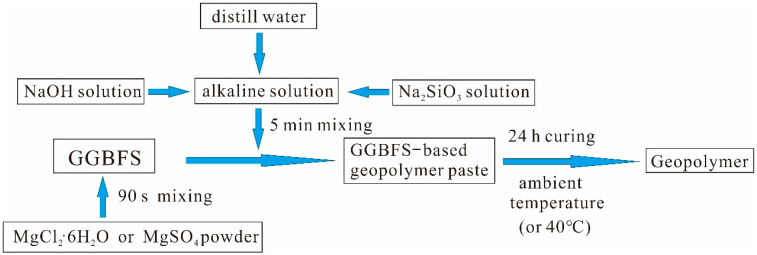
Preparation of GGBFS-based geopolymer.

**Figure 3 materials-15-04911-f003:**
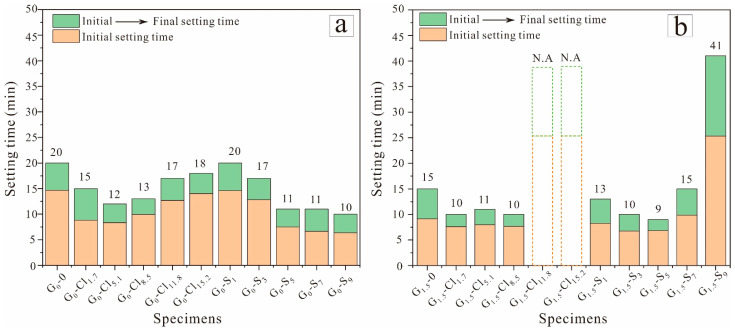
Setting times of (**a**) NaOH-activated and (**b**) Na_2_SiO_3_-activated geopolymers. (Dotted lines indicate specimens that could not set at ambient temperature).

**Figure 4 materials-15-04911-f004:**
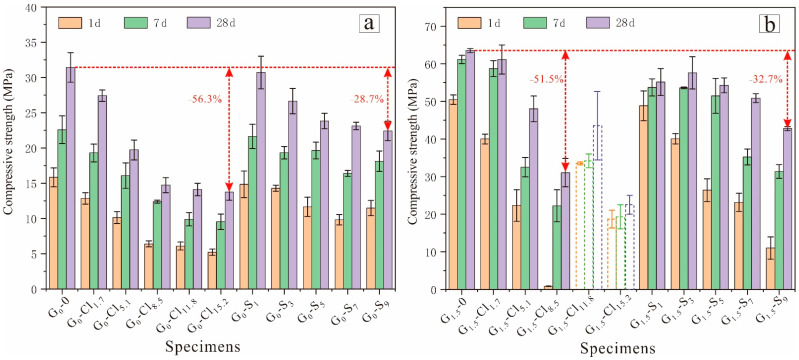
Compressive strengths of (**a**) NaOH-activated and (**b**) Na_2_SiO_3_-activated geopolymers (dotted lines indicate specimens cured at 40 °C).

**Figure 5 materials-15-04911-f005:**
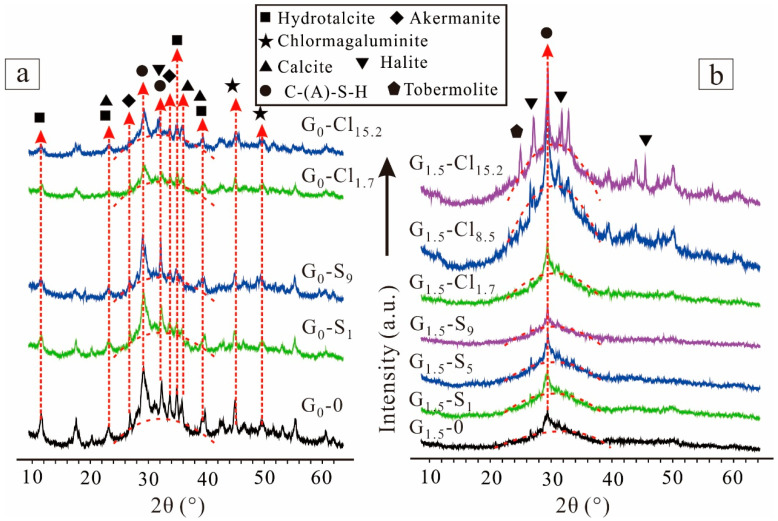
XRD patterns of (**a**) NaOH-activated and (**b**) Na_2_SiO_3_-activated geopolymers.

**Figure 6 materials-15-04911-f006:**
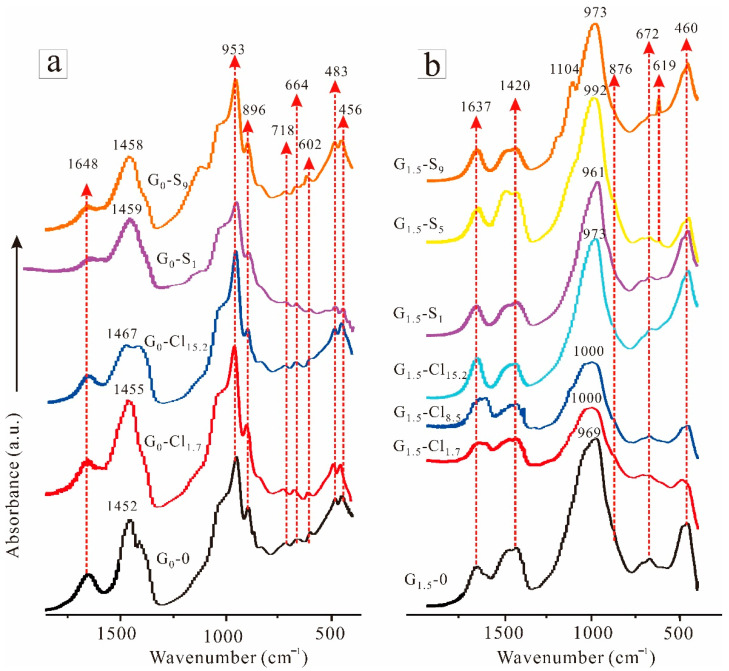
FTIR spectra of (**a**) NaOH-activated and (**b**) Na_2_SiO_3_-activated geopolymers.

**Figure 7 materials-15-04911-f007:**
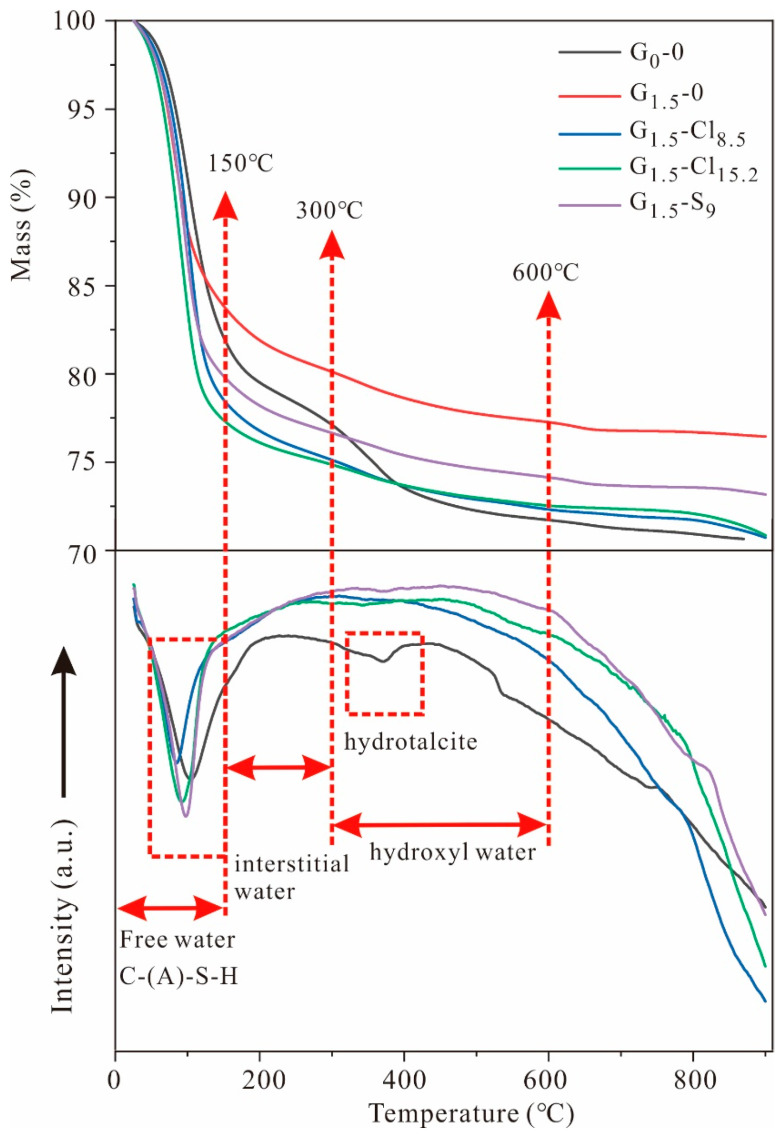
TG and DSC curves of selected geopolymers.

**Figure 8 materials-15-04911-f008:**
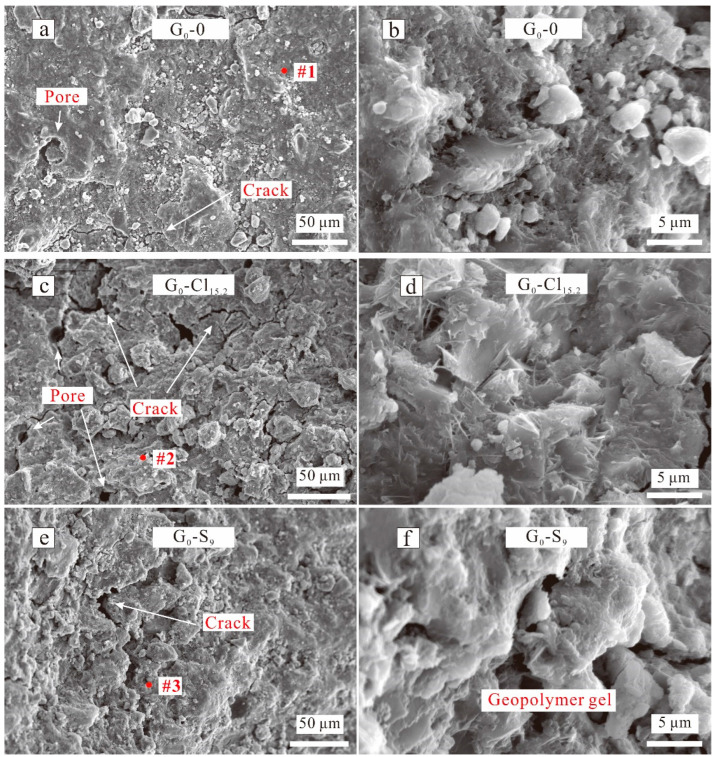
SEM images of NaOH-activated geopolymers of (**a**) G_0_-0, (**c**) G_0_-Cl_15.2_, (**e**) G_0_-S_9_ at low magnification; SEM images of NaOH-activated geopolymers of (**b**) G_0_-0, (**d**) G_0_-Cl_15.2_, (**f**) G_0_-S_9_ at high magnification.

**Figure 9 materials-15-04911-f009:**
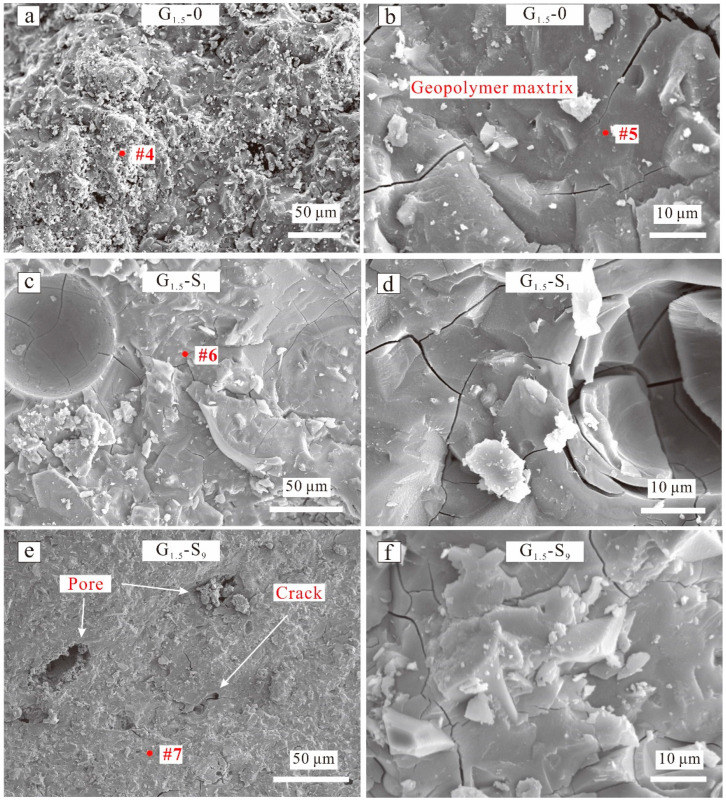
SEM images of Na_2_SiO_4_-activated geopolymers of (**a**) G_1.5_-0, (**c**) G_1.5_-S_1_, (**e**) G_1.5_-S_9_ at low magnification; SEM images of Na_2_SiO_4_-activated geopolymers of (**b**) G_1.5_-0, (**d**) G_1.5_-S_1_, (**f**) G_1.5_-S_9_ at high magnification.

**Figure 10 materials-15-04911-f010:**
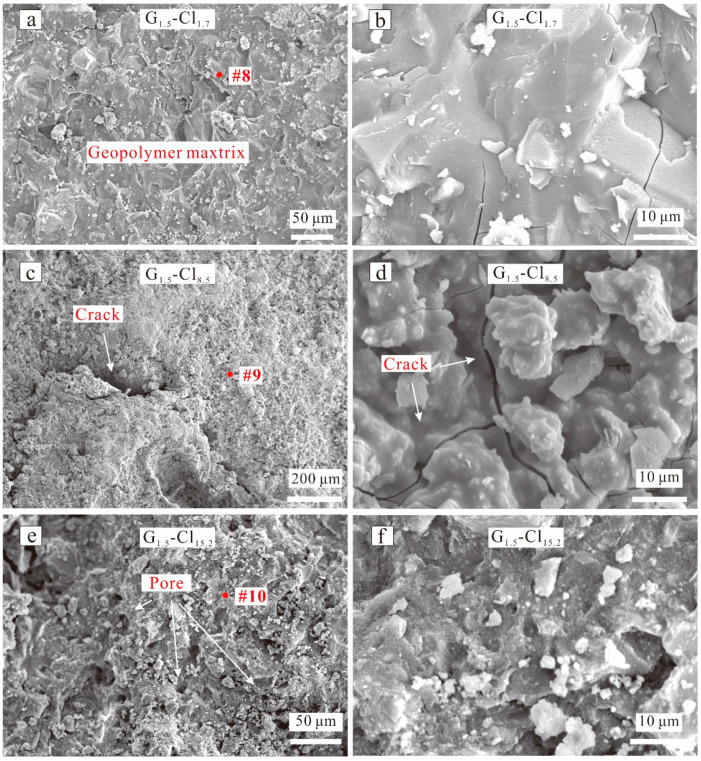
SEM images of Na_2_SiO_4_-activated geopolymers of (**a**) G_1.5_-Cl_1.7_, (**c**) G_1.5_-Cl_8.5_, (**e**) G_1.5_-Cl_15.2_ at low magnification; SEM images of Na_2_SiO_4_-activated geopolymers of (**b**) G_1.5_-Cl_1.7_, (**d**) G_1.5_-Cl_8.5_, (**f**) G_1.5_-Cl_15.2_ at high magnification.

**Figure 11 materials-15-04911-f011:**
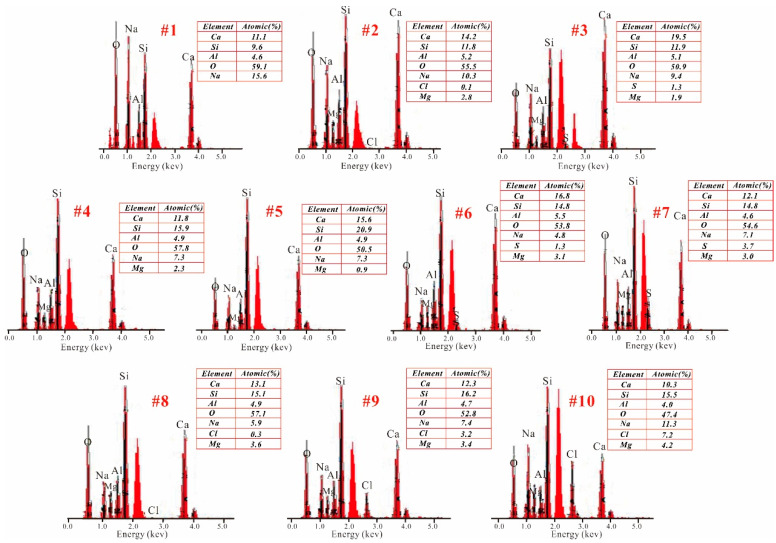
EDX spectra of geopolymers. (The number indicated the red spots in Figure 8, Figure 9 and Figure 10).

**Figure 12 materials-15-04911-f012:**
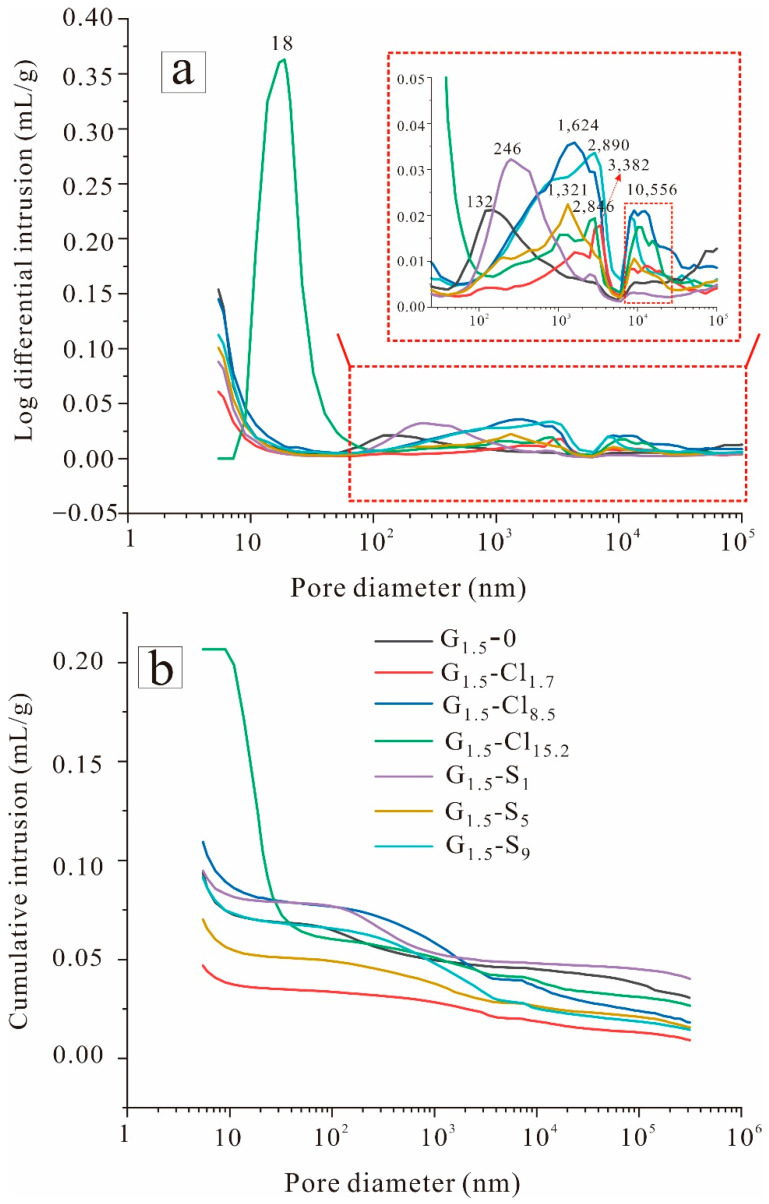
MIP results of geopolymers: (**a**) log differential intrusion and (**b**) cumulative intrusion.

**Table 1 materials-15-04911-t001:** Geopolymer pore parameters based on MIP results.

Samples	Total Pore Area (m^2^/g)	Average Pore Diameter (nm)	Porosity (%)
G_1.5_-0	13.85	26.97	15.81
G_1.5_-Cl_1.7_	6.33	29.65	8.03
G_1.5_-Cl_8.5_	15.48	28.22	16.13
G_1.5_-Cl_18.2_	34.11	24.23	29.25
G_1.5_-S_1_	8.88	42.68	15.41
G_1.5_-S_5_	10.33	27.20	11.51
G_1.5_-S_9_	12.23	29.87	14.45

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
