# Peer review of "Effect of Magnesium Salt (MgCl2 and MgSO4) on the Microstructures and Properties of Ground Granulated Blast Furnace Slag (GGBFS)-Based Geopolymer"

_materials, 2022, doi:10.3390/ma15144911_

Round 1

Reviewer 1 Report

1.      Revise uppercase and lowercase on the title of the manuscript.

2.      Abstract only in one paragraph, revise it.

3.      Quantitative results in the abstract should be added.

4.      Keywords should be reordered based on alphabetical order.

5.      What is the novelty of the present work? Based on my evaluation I do not see something really new presented in the manuscript. GGBFS geopolymers have been widely studied in the past. It needs to explain it more advance.

6.      Section 2 should be changed to “Materials and Methods”.

7.      The experimental workflow needs to be explained as a form of the figure to make the reader more interested and easier to understand rather than only specific figures and text.

8.      Tools information detail such as company manufacturer needs to be included in the manuscript.

9.      Tools accuracy and tolerance should be given.

10.   Discussion in the present manuscript just explains the results without dept with their meaning and influence in the real application. The discussion is mandatorily extended and makes it more concise.

11.   GGBFS have been potentially used in medical applications, such as for cement fixation for the total hip prosthesis. The authors should explain this important point in their introduction and/or discussion section. Also, to support this explanation it is recommended to adopt additional references published by MDPI as follows: Computational Contact Pressure Prediction of CoCrMo, SS 316L and Ti6Al4V Femoral Head against UHMWPE Acetabular Cup under Gait Cycle. J. Funct. Biomater. 2022, 13, 64. https://doi.org/10.3390/jfb13020064

12.   Figure 5 is not clearly seen, and needs more improvement.

13.   The limitation of the present study should be explained clearly.

14.   The conclusion is too long, please make it more concise.

15.   Results in the present study are encouraged to be compared with the previous study.

16.   Further research needs to be explained in the conclusion section.

17.   The authors need to proofread their manuscript to solve grammatical errors and language style issues.

18.   The authors do not use materials, MDPI format correctly. Please use the format after revision.

19.   Overall, this manuscript is still poor, especially in the introduction and discussion section which should be handled by the authors.

Reviewer 2 Report

The paper clearly lacks novelty, scientific soundness, and contribution to the existing knowledge base. The subject area has already been widely investigated and reported in several publications earlier, the closest one among these is “A study on the characteristics and microstructures of GGBS/FA based geopolymer paste and concrete” by Lee et al. (2015) published as Construction and Building Materials Volume 211, 30 June 2019, Pages 807-813, https://doi.org/10.1016/j.conbuildmat.2019.03.291.

Besides, the paper exhibits very high text similarity (23%) with the previously published papers, as identified by the plagiarism analysis by Turnitin and iThenticate. Much of the text is similar to "Effect of curing conditions on the microstructure and mechanical performance of geopolymers derived from nanosized tubular halloysite by Zhang et al. (2021, Construction and Building Materials, Volume 268, 25 January 2021, 121186, https://doi.org/10.1016/j.conbuildmat.2020.121186.

Reviewer 3 Report

The paper could be considered for publication after the following major revision:

1-The English of the paper is not appropriate overall. The language of the paper should be improved.

2-The abstract is long and vague. The authors should clearly state in the abstract what tests and parameters were assess and what the results of those tests were.

3-The introduction of the paper is long. The correlation between microstructure and properties were not well explained. The current format of the introduction is not acceptable.. The following papers should be used to modify this section:

- Effect of melting temperature on microstructural evolutions, behavior and corrosion morphology of Hadfield austenitic manganese steel in the casting process, International Journal of Minerals, Metallurgy, and Materials, Vol. 25 (12), 2018, 1431-1438.

- The effect of pouring temperature and surface angle of vortex casting on microstructural changes and mechanical properties of 7050Al-3 wt% SiC composite, Materials Science and Engineering: A, Vol. 737, 2018, 230-235.

- Optimization of slip casting parameters for spark plasma sintering of transparent MgAl2O4/Si3N4 nanocomposite, Ceramics International, Vol. 45 (16), 2019, 20714-20723.

4-The standard used for sample preparation and compressive strength evaluation was not explain. The repeatability of the results should also be explained.

5-what mechanism(s) was (were) involved for the void and crack formation?

6-The results of figure do not have proper quality. The details of this figure is not clear.

Reviewer 4 Report

The content covered in the publication is very important in terms of reducing the amount of fresh water used in the construction industry. The material, which is a geopolymer due to the many times lower CO2 emissions than commonly used concrete is already a more environmentally friendly material. Detailed knowledge of the mechanisms and changes that occur in the microstructure of the material will have a significant impact on further research directions aimed at reducing fresh water consumption. The authors have already shown in their paper that there are niches in the explanation of some processes. I personally look forward to further publications explaining these processes. 

The topic of the publication corresponds to the content.

The methodology and scope of the research have been correctly defined and the thorough analysis of the results fits perfectly into a scientific publication.

The scope of references is sufficient to correctly foreground the issues described in the publication. 

I recommend the article for publication in its present form.

Round 2

Reviewer 1 Report

The manuscript is recommended for publication.

Reviewer 2 Report

The paper has not been satisfactorily revised as per the reviewer's comments. Since the paper exhibits little novelty and scientific contribution, I can not recommend its publication.

Reviewer 3 Report

The required revision are done. The paper can be published in its current format.